# Inhibition of Hepatic AMPK Pathway Contributes to Free Fatty Acids-Induced Fatty Liver Disease in Laying Hen

**DOI:** 10.3390/metabo12090825

**Published:** 2022-09-01

**Authors:** Cheng Huang, Xiaona Gao, Yan Shi, Lianying Guo, Changming Zhou, Ning Li, Wei Chen, Fan Yang, Guyue Li, Yu Zhuang, Ping Liu, Guoliang Hu, Xiaoquan Guo

**Affiliations:** 1Jiangxi Provincial Key Laboratory for Animal Health, College of Animal Science and Technology, Jiangxi Agricultural University, Nanchang 330045, China; 2School of Computer and Information Engineering, Jiangxi Agricultural University, Nanchang 330045, China

**Keywords:** MAFLD, FFA, AMPK signaling pathway, PCR array, primary hepatocytes

## Abstract

Metabolism-associated fatty liver disease (MAFLD) is one of the most common causes of liver disease; however, the underlying processes remain unknown. This study aimed to investigate the changes of free fatty acids (FFA) on the expression of genes related to the AMP-activated protein kinase (AMPK) signaling pathway in the primary hepatocytes of laying hens. The primary hepatocytes of laying hens were treated with FFA (containing a 2:1 ratio of oleic and palmitic acids) for 24 h. FFA significantly increased lipid droplet accumulation, decreased glycogen synthesis, increased the levels of triglycerides (TG), total cholesterol (TC), reactive oxygen species (ROS), malondialdehyde (MDA), and glucose content in the supernatant (GLU) in the primary hepatocytes of laying hens, and decreased the levels of total antioxidant capacity (T-AOC) and superoxide dismutase (SOD), as well as mitochondrial membrane potential (MMP). The results of the PCR array combined with Western blotting experiments showed that the activity of AMPK was inhibited. Inhibition of AMPK signaling pathway decreases the expression of genes involved in fatty acid oxidation, increases the expression of genes involved in lipid synthesis, decreases the expression of genes involved in glycogen synthesis, increases the expression of genes involved in glycolysis, increases the expression of genes involved in oxidative stress, and increases the expression of genes involved in cell proliferation and apoptosis. Taken together, our results suggest that FFA can affect the homeostasis of the AMPK signaling pathway by altering energy metabolic homeostasis, inducing oxidative stress, and adjusting the onset of cell proliferation and apoptosis.

## 1. Introduction

MAFLD, formerly known as non-alcoholic fatty liver disease (NAFLD), is a condition marked by abnormal lipid metabolism that results in liver steatosis [1,2]. MAFLD progresses from initially benign simple steatosis to non-alcoholic steatohepatitis (NASH), which is characterized by inflammation and ballooning degeneration. In some cases, the disease progresses to various stages of liver fibrosis, cirrhosis, multiple types of hepatic decompensation, and eventually to hepatocellular carcinoma [3,4]. MAFLD is an increasing worldwide health threat that imposes a major economic cost on all countries, with a global prevalence of roughly 25% [5]. It has been discovered that the development of MAFLD may result in an increase in hepatic triglyceride (TGS) accumulation and an increase in the inward flow of free fatty acids (FFAs), affecting intracellular metabolic and signaling pathways in hepatocytes, resulting in oxidative stress, cell cycle disruption, and other adverse effects [6,7]. Despite extensive research, no medicine for the treatment of MAFLD has been approved. As a result, it is critical to continue finding MAFLD therapeutic targets. Although most studies on MAFLD have used rodents as model animals, rodents rely on adipose tissue and the liver for lipid synthesis, whereas humans and chickens similarly rely on the liver for more than 90% of initial lipid production [8]. Additionally, MAFLD may arise in hens during natural aging owing to the impact of estrogen and certain dietary variables [9]. As a consequence, chickens can serve as a great model for investigating human diseases of lipid metabolism.

The AMP-activated protein kinase (AMPK) is a ubiquitously expressed serine/threonine-protein kinase activated in a low cellular energy state [10,11]. Once activated, AMPK triggers a catalytic process to produce ATP while inhibiting anabolic processes that consume ATP to restore cellular energy homeostasis [12]. AMPK leads to the phosphorylation of critical metabolic substrates and transcriptional regulators implicated in almost all branches of cellular metabolism [13]. Thus, AMPK has emerged as an attractive potential candidate for metabolic diseases such as obesity and diabetes mellitus type 2 (T2D). A growing body of evidence has revealed that AMPK is closely associated with the development of MAFLD [14,15].

AMPK and its signaling pathways play a crucial role in regulating cellular energy homeostasis, mitochondrial function, and growth and metabolism [16]. AMPK can shift metabolism to increase catabolism and decrease anabolism through metabolic substrates and transcriptional regulators [17]. For example, AMPK regulates lipids primarily by phosphorylating and inhibiting acetyl coenzyme A carboxylase (ACC), significantly promoting lipid oxidation and inhibiting FA synthesis [18]. By increasing the peroxisome proliferator-activated receptor-gamma coactivator 1-α (PGC-1α), mitochondrial biogenesis is stimulated by AMPK activity, promoting oxidative metabolism [19,20]. AMPK also inhibits anabolic pathways by phosphorylating rapamycin (mTOR) and tuberous sclerosis complex 2 (TSC2) regulatory-associated proteins, thereby causing mTOR inactivate and preventing phosphorylation of its substrates, which is closely related to the regulation of the cell growth cycle [21]. 

Although AMPK plays a central role in regulating multiple aspects of metabolism, our understanding of the role of AMPK in MAFLD has been primarily based on studies conducted on a single part. The overall effect of MAFLD on AMPK and its signaling pathway has been under-explored. Therefore, we used primary hepatocytes from laying hens to construct a steatosis model by adding free fatty acids. The effect of steatosis on AMPK and its signaling pathway in hepatocytes was assessed by using the PCR Array technique, and its potential mechanism was explored. Our study may provide new insights into the role of AMPK in MAFLD interference. 

## 2. Materials and Methods

### 2.1. Cell Isolation and Treatments

The ethics committee approved all experimental procedures involving animals at Jiangxi Agricultural University (JXAULL-2019023). According to our previous study, the culture method of primary hepatocytes from laying hens was established [22]. Primary hepatocytes were housed in six-well plates at an amount of 1 × 10^6^ cells per well. After reaching 70–80% confluency, FFA (a mixture of oleic and palmitic acids in a 2:1 ratio, c/c) was added to the medium at a final concentration of 0 or 1 mM and incubated for 24 h to induce steatosis in primary hepatocytes. The experiments were grouped as follows: 0 mM FFA (CK group) and 1 mMFFA (FFA group) for 24 h, respectively. After treatment, the cells were harvested for further analysis. These doses were chosen based on previous studies and our preliminary results [22,23]. 

### 2.2. Oil Red O and Periodic Acid Schiff Staining

Cells were cultured on slides of 6-well plates and were rinsed three times with PBS; 4% paraformaldehyde was fixed at 4 °C for 15 min, and then cells were rinsed with PBS. Oil Red O reagent (Sigma, Ronkonkoma, NY, USA) was incubated for 30 min, rinsed three times with PBS, stained with hematoxylin for 30 s, and rinsed two times with PBS. Periodic acid solution (Sigma, St. Louis, MO, USA) was oxidized for 5 min. After washing with distilled water several times, the six-well plates were covered and stained for 15 min after adding a sufficient amount of Schiff’s staining solution, rinsed with double-distilled water for 2 min, and then re-stained with Mayer hematoxylin staining solution for 1 min and rinsed with double-distilled water for several times. To determine the hepatocyte staining, the hepatocytes were observed and imaged under a light microscope (Leica, Wetzlar, Germany).

### 2.3. Real-Time Cell Growth Assay

Cell growth was monitored using an iCelligence RTCA instrument (Roche Diagnostics, Shanghai, China). In this assay, 200 μL of the walled medium was added to each well of the E-plate, containing 10,000 cells per well. The system continuously monitors the impedance of the cells for 72 h at the indicated time and measures the value as a ‘cell index’. Data were collected and analyzed using RTCA software 1.2.

### 2.4. Measurement of Cellular Biochemical Indicators

The culture medium and cells were collected at the end of cell treatment. The cellular lipid metabolism-related indexes (TG and TC), cellular supernatant glucose content, cellular antioxidant enzymes, and oxidative stress indexes (SOD, T-AOC, and MDA) were examined in strict accordance with the kit instructions (Nanjing Jiancheng Bioengineering Institute, Nanjing, China).

### 2.5. The Detection of MMP and ROS Level

After cell treatment, the cells were collected with trypsin adjusted to 1 × 10^6^ and centrifuged. The cells were then resuspended with DCFH-DA dye (Nanjing Jiancheng Institute of Biological Engineering, Nanjing, China) at a final concentration of 10 μM. The cells were incubated with DCFH-DA for 30 min at 37 °C, protected from light, and mixed upside down every 5 min to bring the probe and cells into full contact. At the end of incubation, the cells were centrifuged again at 2000 r/min for 5 min, the supernatant was discarded, the cells were washed twice with PBS, and, finally, the cell precipitate was resuspended with PBS while waiting for detection.

The cells were treated and collected in the same manner and MMP was determined using the MMP kit (Beyotime Biotech, Shanghai, China). Then 1 mL of JC-1 staining working solution (1×) was added to the prepared cell suspension and mixed thoroughly. The mixture was then set to incubate in a cell incubator at 37 °C for 20 min. At the end of incubation, the supernatant was aspirated and washed 2 times with JC-1 staining buffer. Cell culture solution was added to resuspend and wait for observation.

Cells were measured and photographed using a flow cytometer (C6 Plus Flow Cytometer, BD, Franklin Lakes, NJ, USA) and an inverted fluorescence microscope (Olympus Optical Co., Ltd., Tokyo, Japan), respectively. Contrast areas with the same number of cells in the pictures were selected, and the average fluorescence intensity of ROS-positive cells in differently treated cells was analyzed using Image J (ImageJ National Institutes of Health, Bethesda, MD, USA). Six samples were repeated for each group, and data were obtained and statistically compared. 

### 2.6. Acridine Orange/Ethidium Bromide (AO/EB) Staining Assessment

Cells were treated with FFA for 24 h, collected, and centrifuged. The number of cells was adjusted to about 1 × 10^6^ cells/mL. Before use, AO (5 μg/mL) solution and EB (5 μg/mL) solution (Keygen Biotech, Nanjing, China) were mixed in a 1:1 ratio to form a working solution, stained according to the instructions of the assay kit, and detected under an inverted fluorescence microscope. The fluorescence intensity was recorded at an excitation wavelength of 510 nm, an emission wavelength of 530 nm for AO, and an emission wavelength of 605 nm for EB. The mean fluorescence intensity was measured using Image J (ImageJ National Institutes of Health, Bethesda, MD, USA).

### 2.7. Pathway Analysis and PCR Array 

We performed an enrichment analysis of 84 genes related to the AMPK signaling pathway using the Metascape platform (http://metascape.org, accessed on 16 December 2021). The enrichment analysis consisted of two main parts: gene ontology (GO) functional enrichment and the Kyoto Encyclopedia of Genes and Genomes (KEGG) pathway enrichment. Functional enrichment analysis, protein-protein interaction (PPI) network, and gene functional clustering network maps were constructed.

After treating primary hepatocytes with FFA for 24 h, total cellular RNA was extracted with TRIzol. RNA samples were assessed quantitatively and qualitatively with an ultra-micro UV spectrophotometer (Quawell Q3000, Thermo Fisher Scientific, Waltham, MA, USA). Using 0.5 µg of total RNA as template, cDNA was synthesized using a reverse transcription kit (Takara, Tokyo, Japan). According to the manufacturer’s protocol, these cDNA products were analyzed using AMPK qPCR arrays (Wcgene Biotech, Shanghai, China). Data were analyzed using Wcgene Biotech software. Genes with fold-changes more than or less than 2.0 were considered biologically significant. The sequences of the primers used in the PCR array analysis are listed in Appendix A.

### 2.8. Validation of PCR Array by Real-Time Quantitative PCR (RT-PCR)

The cell samples were re-assayed following the same model construction and collection methods. We selected some differentially expressed genes in PCR arrays and validated them using qPCR (Quant Studio7, Thermo Fisher Science, Waltham, MA, USA), following the qPCR kit instructions (Takara, Tokyo, Japan).

### 2.9. Western Blot Analysis

Cells were lysed with RIPA lysis buffer (Solarbio Biotechnology Beijing, Beijing, China) supplemented with 1 mM protease and phosphatase inhibitors. Protein concentrations were measured using the BCA Protein Assay Kit (Solarbio Biotechnology Beijing, Beijing, China), separated with 6–15% SDS polyacrylamide gels, and then transferred to polyvinylidene difluoride (PVDF) membranes (Millipore, Billerica, MA, USA). The membranes were blocked with 5% skim milk for 1–2 h at room temperature and incubated at 4 °C overnight with different primary antibodies: AMPK (1:1000, Wanleibio, Shenyang, China, WL02254), P-AMPK (Thr 172) (1:2000, Cell Signaling Technology, Danvers, MA, USA, #2531), P-ACC (Ser79) (1:2000, Cell Signaling Technology, Danvers, MA, USA, #3661), ACC (1:2000, Cell Signaling Technology, Danvers, MA, USA, #3662), AKT (1:1000, Abcam, Cambridge, MA, USA), ab38449), P-AKT (Ser473) (1:1000, Wanleibio, Shenyang, China, WLP001a), Insulin Receptor (INSR) (1:500, Wanleibio, Shenyang, China, WL02857), and GAPDH (1:2000, Cell Signaling Technology, Danvers, MA, USA, #5174S). The washed membranes were incubated with the corresponding secondary antibodies for 1–2 h. Finally, the signal was detected with bio rad Chemidoc Touch imager (Bio-Rad Chemidoc Touch, Hercules, CA, USA) using enhanced chemiluminescence kits (ECL, vazyme, Nanjing, China).

### 2.10. Statistical Analysis

The current results were derived from at least three independent experiments. All data were processed with SPSS 25.0 software (Version 25; IBM, Armonk, NY, USA) and expressed as mean ± SD. Independent samples *t*-test was used for statistical analysis. Finally, data were presented using GraphPad Prism 6 software (Version 6; La Jolla, CA, USA). 

The results of PCR arrays were analyzed, and differentially expressed genes (DEG) heat maps were created using R software. A volcano map was created with the R software packages “ggplot2” and “ggrepel” to analyze the differences between the groups. The significant difference was declared when *p* < 0.05.

## 3. Results

### 3.1. FFA Induced Lipid Accumulation and Reduced Glycogen Synthesis 

Compared with the CK group, primary hepatocytes incubated with 1 mM of FFA showed steatosis and time-dependent accumulation of intracellular lipid droplets, but the nuclei were visible (Appendix A). Oil Red O staining showed a large number of red lipid droplets or even aggregation of lipid droplets in the cytoplasm of the cells in the FFA group, and the nuclei were blue (Figure 1A). In addition, FFA induced elevated TG and TC levels in hepatocytes (Figure 1C,D).

Glucose uptake and glycogen synthesis are two of the most critical indicators in the study of glucose metabolism. The PAS staining results showed that the glycogen synthesis of the cells in the FFA group was inhibited. The number of intracellular glycogen-positive compartments was reduced, and the staining intensity was decreased (Figure 1B). Moreover, the ability of cells in the FFA group to consume glucose was extremely significantly reduced compared to the CK group (*p* < 0.01) (Figure 1E).

### 3.2. FFA Induces Excessive ROS Production and Oxidative Stress

After FFA treatment of cells, ROS levels were detected at 24 h (Figure 2A–D). The results showed that the intracellular ROS fluorescence intensity was significantly higher in the FFA group (*p* < 0.01) compared with the CK group. As shown in Figure 2E–G, SOD, T-AOC, and MDA levels were markedly higher in the FFA group compared with the CK group (*p* < 0.01).

### 3.3. FFA Increases Apoptosis and Inhibits Proliferation

The proliferation of hepatocytes was detected in real-time with the iCelligence system and E-plate. As shown in Figure 3A, the cells in the FFA group grew faster than those in the CK group for 12 h. After that, the cell growth curve clearly showed that the cells in the FFA group grew significantly slower than the cells in the CK group. The experimental results indicate that at a later stage of FFA treatment, cell proliferation is delayed.

Cell growth may be halted in conjunction with the activation of apoptosis. Disruption of the mitochondrial transmembrane potential is one of the early intracellular processes associated with the initiation of apoptosis. Apoptosis was detected with flow cytometry. JC-1 exists in both monomeric and multimeric states. In normal cells, where the MMP is normal, JC-1 accumulates in mitochondria as a multimer, emits red fluorescence, and is usually detected as a PE channel. In apoptotic cells, the mitochondrial transmembrane potential is depolarized, and JC-1 is released from the mitochondria in a reduced concentration, reversing to the monomeric form, emitting green fluorescence, usually a FITC channel when detected. Thus, the apoptotic cell population (low MMP) is reflected in the high FITC and low PE quadrants. The results (Figure 3B,C) showed that the percentage of apoptotic cells in the FFA group was significantly higher than that of the CK group (*p* < 0.01). To further confirm the effect of FFA on apoptosis, we examined this effect with AO/EB staining. Necrotic cells appeared red after staining, while apoptotic cells emitted bright orange fluorescence, and green fluorescence represented normal cells. As shown in Figure 3D, many orange and red cells were observed in the FFA group. And the number of bright orange cells was less in the CK group. The statistical results in Figure 3E further demonstrated that apoptotic cells were significantly increased in the FFA group compared with the CK group (*p* < 0.01).

### 3.4. Functional Enrichment Analysis of AMPK Signaling Pathway

We added genes into Metascape to examine the AMPK signaling pathway and its possible mode of action. The analysis results are shown in Figure 4A,C, as based on the results of functional enrichment analysis. AMPK signaling pathways were mainly associated with biological processes such as regulation of protein kinase activity, carbohydrate metabolic processes, glucose homeostasis, positive regulation of cellular catabolic processes, sterol biosynthesis processes, cellular response to fatty acids, cellular response to decreased oxygen level, and autophagy of mitochondrial. The PPI network was mainly focused on genes such as PRKAR, STRAD, SREBF1, AKT, ATG13, MTOR, and EEF2K (Figure 4B). These results suggested that the AMPK signaling pathway was closely related to glucolipid metabolism, mitochondrial homeostasis, and redox, which prompted us to elucidate the potential relationship between AMPK and hepatocyte steatosis comprehensively.

### 3.5. Effect of FFA on Genes Related to AMPK Signaling Pathway

Previously, we have shown that FFA could induce lipid accumulation and reduced glycogen synthesis in the primary hepatocytes of laying hens and cause oxidative stress, altering the onset of cell proliferation and apoptosis. Given that AMPK plays a crucial role in regulating energy homeostasis and cellular metabolism, we further conducted an experimental study on the effect of adding FFA on the AMPK signaling pathway. We first detected the changes in AMPK-related gene expression with a PCR array kit. As shown in Figure 5A,B, several genes in the CK and FFA groups showed significant differences after modeling. In comparison to the CK group, the expression of genes involved in lipid metabolism (fatty acid synthase (FASN), Acetyl-CoA Carboxylase Beta (ACACB), adrenoceptor alpha 2A (ADRA2A), adrenoceptor alpha 2C (ADRA2C), 3-Hydroxy-3-Methylglutaryl-CoA Synthase 1 (HMGCS1)), genes related to glucose metabolism (6-Phosphofructo-2-Kinase/Fructose-2,6-Biphosphatase 3 (PFKFB3), 6-Phosphofructo-2-Kinase/Fructose-2,6-Bisphosphatase 4 (PFKFB4)), genes related to oxidative stress (Thioredoxin Interacting Protein (TXNIP), Forkhead Box O3 (FOXO3)), and genes related to cell proliferation and apoptosis (mTOR, Tumor Protein P53 (TP53), TSC Complex Subunit 1 (TSC1), TSC2) were significantly up-regulated, and the expression of genes related to lipid metabolism (CREB Regulated Transcription Coactivator 1(CRTC1), Carnitine Palmitoyl transferase 2 (CPT2), CPT1A), genes related to glucose metabolism (INSR, PPARG Coactivator 1 Beta (PPARGC1B)), genes related to oxidative stress (Calcium/Calmodulin Dependent Protein Kinase 2 (CAMKK2)), and genes related to cell proliferation and apoptosis (AKT Serine/Threonine Kinase 1 (AKT1)) were significantly down-regulated. RT-PCR examined eight genes to validate the results of PCR arrays. Differences between genes were represented with curve fitting, which indicated that RT-PCR results were generally consistent with PCR array results (Appendix A).

### 3.6. Translational Level Detection of the Effect of FFA on Hepatocytes

Further, to clarify the regulation effect of FFA on AMPK signaling pathway, the Western blot was used to investigate the protein expressions of the AMPK signaling pathway. The results were shown in Figure 6A–F compared with the CK group, the P-AMPK and P-ACC protein levels were significantly lower in the FFA group (*p* < 0.01); ACC protein levels were significantly higher (*p* < 0.05). AKT and P-AKT protein expression showed a decreasing trend, but the difference was not significant (*p* > 0.05); conversely, INSR protein expression showed a decreasing trend, but the difference was not significant (*p* > 0.05).

## 4. Discussion

MAFLD is a primary cause of non-communicable disease mortality in laying hens, resulting in significant economic losses in poultry production [24,25]. Additionally, chickens and people have comparable lipid synthesis pathways, making chickens an ideal model for human MAFLD research. Nevertheless, the actual cause of MAFLD in laying hens is unknown, and no specific target genes have been identified [26]. Notably, AMPK is regarded to be critical for maintaining cellular energy balance and may function as a key protein in a variety of signaling pathways [12]. Previous research has demonstrated that variations in the AMP/ATP ratio during the development of MAFLD in people have an effect on the activation of AMPK [27]. However, the role of AMPK in MAFLD in laying hens has not been studied in depth. In recent years, there have been numerous reports on chicken hepatocytes as a study subject for MAFLD, and most of the sources for chicken hepatocytes were obtained with chicks extraction [8,28]. Moreover, since the onset of MAFLD in laying hens occurs at the peak of egg production, the use of FFA-treated primary hepatocytes from adult laying hens is more representative of the process of MAFLD than that of chicks [29]. As such, we sought to determine the involvement of the AMPK signaling pathway in controlling steatosis in primary hepatocytes from laying hens in order to provide some support for the prevention and treatment of human MAFLD.

By far the most crucial stage in the progression of MAFLD is an increase in hepatic free fatty acid concentration [30]. Additional fatty acid processing occurs in the synthesis of TG and TC storage, which increases metabolic load and lipotoxicity in hepatocytes. Additionally, several studies have shown that treating hepatocytes with FFAs might result in lipid buildup and disruption of lipid metabolism [31,32]. As a consequence, we assessed the effect of FFA-induced hepatocyte steatosis. The findings of this study demonstrated that cells treated with free fatty acids produced more lipid droplets and had elevated levels of TG and TC. In the majority of instances, aberrant lipid deposition is closely related to the balance of adipogenesis and lipolysis and is regulated by lipid metabolism factors. It has been proven that dissociating AMPK from its upstream kinase LKB1 decreases AMPK threonine 172 phosphorylation, resulting in lower AMPK activity, upregulation of ACAC and FASN expression, and downregulation of CPT-1A expression in a nutrient-rich environment [33,34]. We demonstrated in this study that FFA dramatically affected the expression of AMPK and lipid metabolism-related factors in hepatocytes and observed that fatty acid oxidation-related genes (CRTC1, CPT1A, and CPT2) were downregulated, whilst lipid synthesis-related genes (FASN, ACACB, ADRA2A, ADRA2C, and HMGCS1) were upregulated. By regulating the expression of lipid metabolism-related proteins, activating AMPK has been shown to relieve free fatty acid-induced MAFLD in vivo and in vitro [35]. To conclude, FFA induces steatosis in hepatocytes and inhibits AMPK activity, increasing lipid synthesis and decreasing fatty acid oxidation.

Excess FFA is regarded as a risk factor for the development of insulin resistance. In gluconeogenesis, glucose absorption and gluconeogenesis are critical metabolic markers, and glycogen synthesis helps maintain the dynamic balance of sugars [36]. It has been proven that diabetic individuals’ liver glycogen levels are decreased [37,38]. The PAS and glucose levels of the supernatant during this experiment also indicated that the FFA group’s hepatocytes had a much lower glycogen content than the CK group. As a result, we hypothesized that this was the outcome of insulin resistance, which results in decreased glycogen synthesis and lower glucose consumption. It was discovered that AMPK signaling pathways are critical for glucose homeostasis regulation, and that their inactivation results in hepatic insulin resistance [39]. We detected a decrease in the expression of the INSR/AKT/AMPK signaling cascade. Although in our study, P-AKT did not realize a significant decrease, there was a definite downward trend, probably due to the expression of the gene at the translational level lagging behind the transcriptional level. It indicates that FFA promotes gluconeogenesis and inhibits gluconeogenesis, altering the balance of glucose metabolism. Increased AMPK, INSR, and AKT phosphorylation has been shown to enhance insulin sensitivity in the liver by inhibiting gluconeogenesis and increasing glucose absorption [40]. Additionally, AMPK participates in glycolysis via PFKFB3. We noticed a substantial increase in PFKFB3 and PFKFB4 mRNA expression in the FFA group in this study. Notably, increased PFKFB3 expression and glycolysis are early indications of activation of hepatic stellate cells and are associated with higher expression of fibrosis markers [41]. We hypothesize that steatosis may result in hepatocyte fibrosis following steatosis, a condition that resembles the development of MAFLD. The findings above imply that the addition of FFA affected the expression of gluconeogenic genes in the AMPK signaling pathway, hence decreasing hepatocyte glycogen production and disturbing the gluconeogenic balance.

Steatosis in hepatocytes is intimately tied to oxidative stress. Disruptions in lipid metabolism result in increased oxidation, which eventually results in elevated ROS levels [42]. ROS is thought to be a critical regulator of oxidative stress, which is often generated by an imbalance of pro-oxidants and antioxidants [43]. According to our present findings, the addition of FFA increased the buildup of ROS and MDA and reduced the activity of SOD and T-AOC. Numerous studies have demonstrated that cells lacking AMPK activation produce excessive ROS and induce oxidative stress [19,44]. Subsequently, we examined the AMPK signaling pathway following FFA addition and discovered that FFA lowered AMPK activity while increasing FOXO3 and TXNIP expression. Metformin, an AMPK activator, was shown to inhibit FOXO3 and reduce cellular ROS levels. TXNIP is a redox switch that enhances oxidative stress in cells by inhibiting thioredoxin (TRX) and lowering oxidant scavenging capacity [45]. Furthermore, FOXO3 is a transcription factor upstream of TXNIP that induces its transcription. As a result, we reasoned that steatosis increased the generation of ROS in hepatocytes, changed the expression of the AMPK/FOXO3/TXNIP signaling cascade, created an imbalance between pro-oxidants and antioxidants, and resulted in oxidative stress.

The buildup of free fatty acids results in an abnormal accumulation of ROS in hepatocytes, and oxidative stress can induce apoptosis, which has an effect on the development of MAFLD [34]. The addition of FFA dramatically decreased hepatocyte proliferation, increased MDA content, decreased cell MMP, and increased apoptotic cells. These findings imply that FFA has an influence on hepatocyte growth and apoptosis. Numerous regulatory variables are known to affect cell growth and death. Increased AMPK phosphorylation has been demonstrated to prevent apoptosis in hepatocytes, and liver-specific AMPK deficiency exacerbates liver damage in a mouse model of NASH [46]. Notably, the relationship between the AMPK pathway and apoptosis and proliferation in the etiology of MAFLD remains unknown. To this goal, we investigated the AMPK signaling pathway’s impact on apoptosis and proliferation blockade. FFA treatment lowered AMPK activity while increasing TP53, TSC1, TSC2, and mTOR expression in hepatocytes. Derdak et al. showed that AMPK regulates cell proliferation and death mostly through the TP53-TSC2-mTOR pathway [47]. TP53, an upstream gene of AMPK, detects a variety of stress signals and initiates cell proliferation arrest, senescence, or apoptosis, and activation of TP53 can induce TSC1/2 expression. Additionally, AMPK acts as a negative regulator of TSC1/2-mTOR signaling, and AMPK deficiency results in mTOR overexpression [48]. Additionally, it has been observed that activating AMPK and delaying TP53 activation with exercise can suppress hepatocyte proliferation and attenuate steatosis and liver damage in an obese mouse model [49]. Steatosis, in conclusion, inhibits hepatocyte growth and induces apoptosis.

In addition to this, it is widely recognized that the use of the activators of AMPK can improve NASH by promoting AMPK phosphorylation, reducing hepatic lipid metabolism disorders, decreasing inflammation, and attenuating fibrosis in in vivo and in vitro trials [50]. One study found that, in mice, metformin activated AMPK downstream signaling and reduced hepatic lipid accumulation [51]. Treponelactone reduces hepatic lipid deposition, inflammation, and fibrosis in NASH patients, and has the potential to reduce NAFLD as a metabotropic AMPK agonist [52]. This also reminds us of the therapeutic role of increasing AMPK activity in subsequent studies for MAFLD in laying hens.

## 5. Conclusions

In conclusion, FFA can induce disruption of lipid metabolism and reduction of glycogen synthesis by altering the AMPK signaling pathway in primary laying hen hepatocytes, causing oxidative stress, contributing to cell proliferation arrest, and inducing the onset of apoptosis. This lays the foundation for further study of the inhibitory mechanism of MAFLD on the AMPK signaling pathway.

## Figures and Tables

**Figure 1 metabolites-12-00825-f001:**
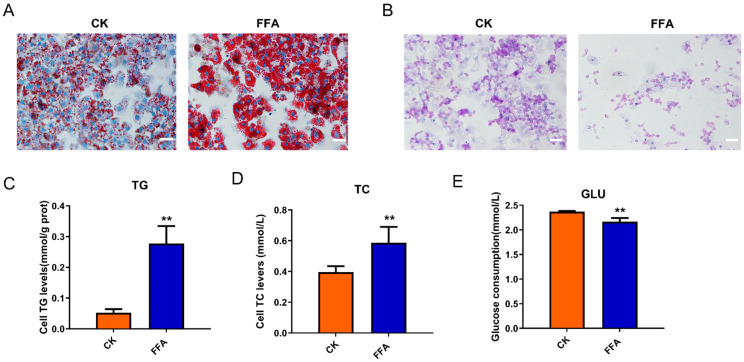
Effect of FFA treatment for 24 h on glycolipid metabolism-related indexes in primary hepatocytes of laying hens. (**A**) Cells from each experimental group were stained with Oil Red O. (**B**) Cells from each experimental group were stained with PAS. The scale bar represents 50 μm (**C**) Cellular triglyceride levels of each experimental group. (**D**) A total cholesterol level of cells in each experimental group. (**E**) Levels of cell supernatant glucose. Data are shown as the mean ± SD (n = 6), with (**) *p* < 0.01.

**Figure 2 metabolites-12-00825-f002:**
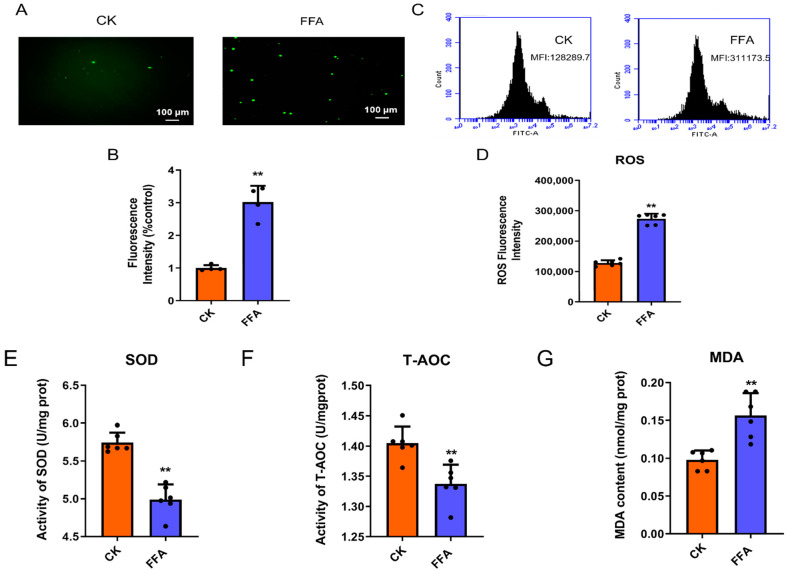
Effect of FFA treatment for 24 h on oxidative stress in primary hepatocytes of laying hens. (**A**–**D**) Effect of FFA on intracellular ROS levels (**A**): inverted fluorescence microscopy; (**B**): Average fluorescence intensity of ROS; (**C**,**D**): flow cytometry. (**E**) Cellular superoxide dismutase activity in each experimental group. (**F**) Cellular malondialdehyde content in each experimental group. (**G**) Total antioxidant capacity of cells in each experimental group. Data are shown as the mean ± SD (n = 6), with (**) *p* < 0.01.

**Figure 3 metabolites-12-00825-f003:**
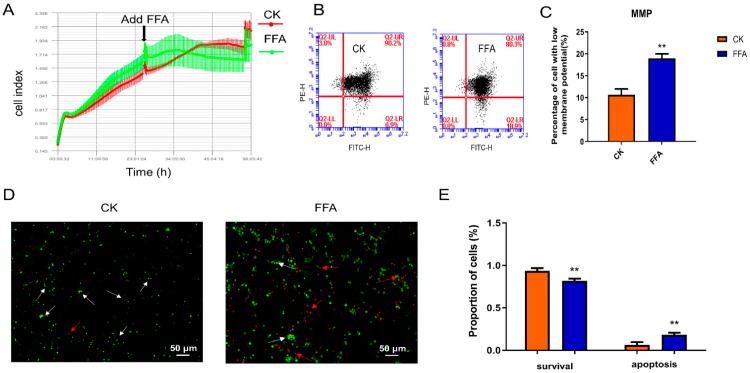
Effect of FFA treatment for 24 h on proliferation and apoptosis of primary hepatocytes in laying hens. (**A**) The cell growth rate was monitored with an iCelligence real-time monitoring system. (**B**) Flow cytometry analysis of MMP. (**C**) The plot of MMP changes. (**D**) AO/EB staining and fluorescence microscopy analysis. (**E**) Data are expressed as the percentage of apoptotic cells in histogram statistics. White arrows point to normal cells and red arrows point to apoptotic cells. Data are shown as the mean ± SD (n = 6), with (**) *p* < 0.01.

**Figure 4 metabolites-12-00825-f004:**
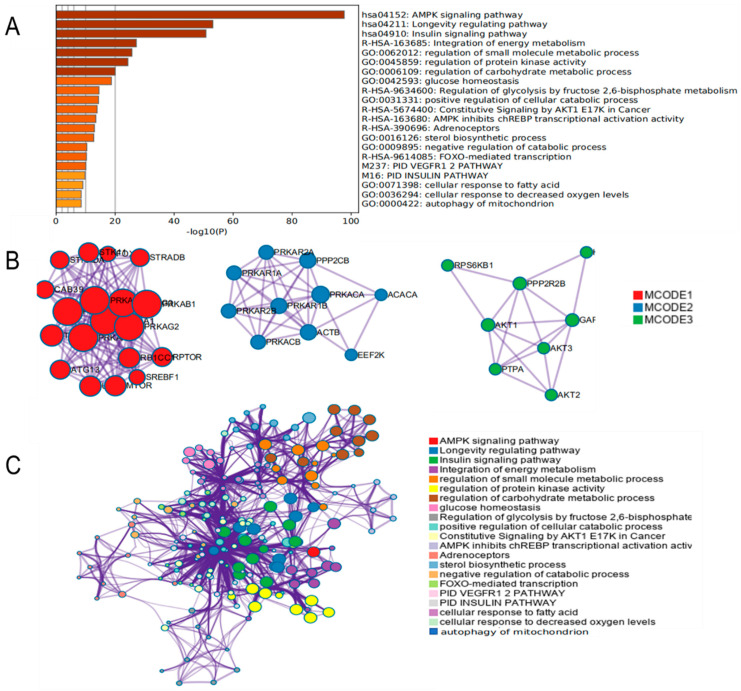
Functional enrichment analysis and protein interactions analysis of AMPK signaling pathway using Metascape. (**A**) Histogram of gene functional enrichment analysis. (**B**) Protein interaction analysis graph. (**C**) Network diagram of gene functional enrichment analysis.

**Figure 5 metabolites-12-00825-f005:**
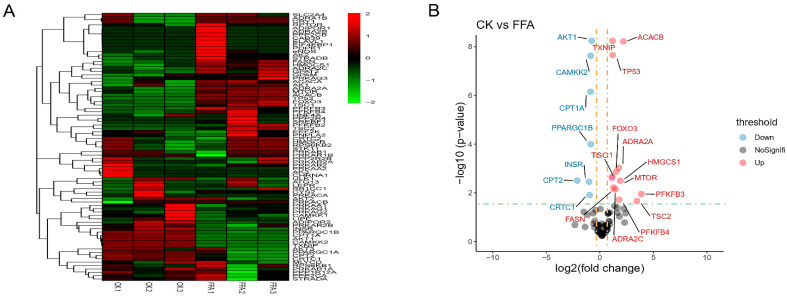
Effect of FFA treatment for 24 h on the expression of genes related to the AMPK signaling pathway in primary hepatocytes of laying hens. (**A**) Heatmap. The color indicates the gene expression level of log10 transformation (red indicates high expression level, green indicates low expression level). (**B**) Volcano plot. Gray dots indicate genes with no significant expression differences between groups under the dashed line, and colored dots above the dashed line indicate genes with significant expression differences between groups. Red indicates increased expression, while blue indicates decreased expression. Data are shown as the mean ± SD (n = 6).

**Figure 6 metabolites-12-00825-f006:**
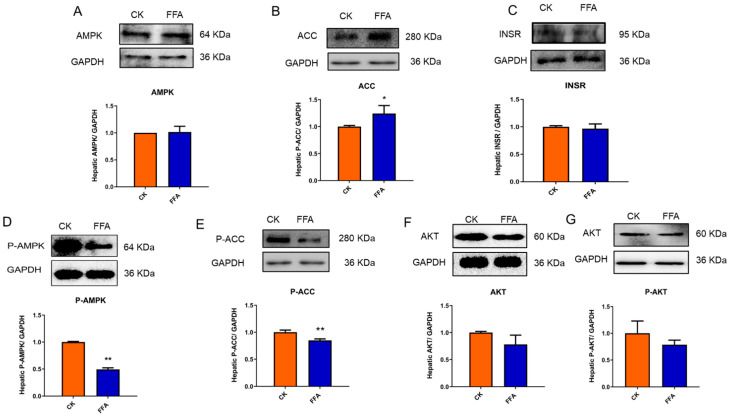
Effect of FFA treatment for 24 h on protein expression levels of genes related to primary hepatocytes in laying hens. (**A**–**G**) Protein expression levels of AMPK, ACC, INSR, p-AMPK, p-ACC, AKT, and P-AKT. Data are shown as the mean ± SD (n = 3), with (*) *p* < 0.05, (**) *p* < 0.01.

## Data Availability

The data presented in this study are available in article and Appendix A.

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
