# Peer review of "Inhibition of Hepatic AMPK Pathway Contributes to Free Fatty Acids-Induced Fatty Liver Disease in Laying Hen"

_metabolites, 2022, doi:10.3390/metabo12090825_

Round 1

Reviewer 1 Report

     The manuscript titled “Correlation of the AMPK signaling pathway with FFA-induced MAFLD in laying hen primary hepatocytes” by Huang et al., is interesting. The authors showed the connection between free fatty acid accumulation, inhibition of AMPK and development of MAFLD. Authors have provided enough evidence regarding the accumulation of free fatty acid and metabolic changes leading to the development of hepatocyte steatosis and MAFLD. Further, they were able to successfully link these changes to AMPK regulatory circuit. It would have been nice if they could have provided some actual therapeutic evidence for the use of AMPK activator in the treatment of MAFLD or at least discussed this point in discussion section. I have some comments in the manuscript that are listed below.

1.     The AMPK signaling pathway decreased the expression of genes involved in fatty acid oxidation, increased the expression of genes involved in lipid synthesis, decreased the expression of genes involved in glycogen synthesis, increased the expression of genes involved in glycolysis, increased the expression of genes involved in oxidative stress, and increased the expression of genes involved in cell proliferation and apoptosis”.  This sentence is not very clear. What does the author mean? Whether they wanted to say that the inhibition of AMPK signaling pathway leads to decreased expression of genes involved in fatty acid oxidation……..”

2.     Figure 1B citation is missing.

3.     Figure Legends for 1A and 1B seems to be wrongly cited. In the text, Figure 1A talks about Oil Red O staining however in the Figure Legend it is cited as Figure 1B and so on.

4.     Please show individual points on the graph in Figure 1C-E; Figure 2C-F.

5.     Please explain clearly Figure 1B, if possible, please keep the cell crawl slide for both condition for similar cell density.

6.     Please provide the quantitation of Figure 2 A. 

7.     Section 3.4 Functional enrichment analysis of AMPK signaling pathway is not very clear. It is difficult to understand about which genes the authors are talking about. Please expand it.

8.     Have the authors investigated the levels of phospho AKT? Generally, AMPK activity is required for AKT phosphorylation rather than AKT total levels.

9.     Is there any evidence that increasing AMPK activity using AMPK activators could delay hepatocyte steatosis and MAFLD? This issue should be discussed in the discussion.

10.  Authors should add a small paragraph highlighting the possible therapeutic opportunity of the use of AMPK activators for MAFLD.

Reviewer 2 Report

the manuscript is well-presented

Reviewer 3 Report

In this manuscript, Xiaoquan Guo’s group continued their recently published work (https://doi.org/10.1016/j.psj.2020.10.059) and further characterized FFA-treated primary hepatocytes of hens. The presented data indicates that FFAs induce lipid accumulation, perturbed cell proliferation, and increased apoptosis via decreased AMPK activity in the hepatocytes.

Major points:

The reviewer finds it hard to appreciate the significance of the reduced AMPK activity in MAFLD without genetics or pharmacological data. The rationale for focusing on the AMPK pathway that the authors provided, the role of AMPK in MAFLD in hens has not been studied in depth, is rather weak. AMPK is a kinase complex, and its signaling is mainly mediated by the cascades of phosphorylation; therefore, the western blot data are essential. Unfortunately, some of the blots have too much noise and background, making it hard to interpret the data, e.g., INSR and p-AMPK. Moreover, the orders of the samples in the western blots are inconsistent, e.g., ACC and p-ACC, and the blots of GAPDH loading controls were not provided. The main issue, though, is an ethical one. The practice of picking/selecting different sets (bands) of biological replicates from the western blots may be misleading. For example, CK1 is presented for the ACC expression level, while CK3 was chosen for the p-ACC level.     

The reviewer also notes that the AMPK pathway was thoroughly investigated in FFA-treated human HepG2 and murine primary hepatocytes as a hepatic steatosis model by Yi Zhang’s group earlier this year (https://doi.org/10.1016/j.metabol.2022.155200), and the apoptosis-mediated lipotoxicity of FAA-treated HepG2 has been documented elsewhere (PMID: 21654881).

Minor points:

The use of MAFLD in the title of the manuscript may not be appropriate as the primary hepatocytes were not derived from hens with MAFLD but isolated from the healthy animals.  

Line 90. 106 should be 106

Line 90-91. Cell fusion. Confluency may be a better choice.

Line 112. A space after 72

Line 121-125. Expand and explain the experiments in more detail. For example, what are these kits measuring, and how the cell numbers were counted with the fluorescence microscope?

Line 126. Spell out AO and EB

Line 127 and 128. What were the concentrations of AO and EB?

Line 130. No excitation with 475 nm?

Line 132. How were the 84 genes chosen?

Line 161. Spell out INSR.

Figure 1A and 1B. Correct the figure legend. 1A should be H&E and oil red O staining, while 1B should be PAS.

Figure 2A. No need for two sets of images. One set is sufficient.

Figure 2C. Correct the Y-axis label. Should not be % control.

Figure 2D. Correct the Y-axis label.

Figure 2E. Make sure the unit of the activity is correct.

Figure 3A. Add h to the X-axis label.

Figure 3B. What was measured by PE and FITC?

Figure 3D. What are the white and red arrows for? Why are there more cells in FFA?

Figure 3E. Remove “the” from the Y-axis label. The ratio should be EB over AO. How many cells were counted, and how?

Line 228. Is it iCelligence or x? Keep it consistent.

Figure 4B. The labels of the nodes are not legible.

Line 266. Remove the space before Insulin.

Figure 5C and 5D. How were the eight transcripts chosen? These data do not add much value and can be moved to supplementary data.

Figure 6D and 6E. Provide the phosphorylation site.

Figure 6F. Consider providing phosphor-AKT

Line 372. MMP

Line 373. Where was Caspase 3 data? 

Author Response

请参阅附件。

Round 2

Reviewer 3 Report

Minor points

Line 128: The reviewer did not understand the meaning of “blown and mixed upside down” in the context.

Line 140: State the number of cells analyzed with Image J, and describe how the measurement was done.

Line 151: Include emission wavelengths for AO and EB.

Line 215: The reviewer did not understand the meaning of “cell crawl sections.”

Line 244: Include the description of the JC-1 staining assay and indicates that the apoptotic cell population (low MMP) reflects in the high FITC and low PE quadrant.

Figure 1. The reviewer suggests changing the blue color of the bar graphs into either light blue or increasing the transparency. The individual data points cannot be appreciated in the present form.

Figure 3D. In the figure legend, describe what the white and red arrows are pointing to. The reviewer finds some white arrows do not point to any cells. Double check the arrows are placed in the right locations.

As stated in the previous feedback, Figure 5C in the revised manuscript does not add much value, and the reviewer recommends moving it to the supplementary data. The reviewer also did not find Figure 5C from the original manuscript in the supplementary data. Please ensure it is transferred.

In the Discussion section, briefly interpret the meaning of the western data where p-AKT was not changed. 
